# Predicting the global mammalian viral sharing network using phylogeography

Gregory F. Albery ⬤ [1,2,3✉], Evan A. Eskew ⬤ [1], Noam Ross ⬤ [1] & Kevin J. Olival ⬤ [1✉]

Understanding interspecific viral transmission is key to understanding viral ecology and evolution, disease spillover into humans, and the consequences of global change. Prior studies have uncovered macroecological drivers of viral sharing, but analyses have never attempted to predict viral sharing in a pan-mammalian context. Using a conservative modelling framework, we confirm that host phylogenetic similarity and geographic range overlap are strong, nonlinear predictors of viral sharing among species across the entire mammal class. Using these traits, we predict global viral sharing patterns of 4196 mammal species and show that our simulated network successfully predicts viral sharing and reservoir host status using internal validation and an external dataset. We predict high rates of mammalian viral sharing in the tropics, particularly among rodents and bats, and within- and between-order sharing differed geographically and taxonomically. Our results emphasize the importance of ecological and phylogenetic factors in shaping mammalian viral communities, and provide a robust, general model to predict viral host range and guide pathogen surveillance and conservation efforts.

[1] EcoHealth Alliance, New York, NY, USA. [2] Institute of Evolutionary Biology, University of Edinburgh, Edinburgh, Scotland. [3] Department of Biology, Georgetown University, Washington, DC, USA. ✉email: gfalbery@gmail.com; olival@ecohealthalliance.org

 1

Most emerging human viruses originate in wild mammals, so understanding the drivers of interspecific viral transmission in these taxa is an important public health research priority[1,2]. Despite a rapidly expanding knowledge base, the mammalian viruses known to science remain taxonomically biased and limited in scope, likely comprising less than 1% of the complete mammalian virome[3,4]. Furthermore, host range is inadequately characterised even for the best-studied viruses[5–7]. To help prioritise viral discovery efforts and zoonotic disease surveillance in wildlife, studies have revealed high (zoonotic) parasite diversity in certain host taxa, such as rodents and bats[5,8], and/or linked parasite diversity with host phenotypic traits such as reproductive output[9,10]. Viral diversity has also been associated with host macroecological traits, including geographic range size[11] and sympatry with other mammals[5]. The rationale for investigating viral diversity is that species with more viruses will generate more opportunities for viral transmission to other species, including humans. However, in order to infect a new host species, a virus must transmit, invade, and then replicate within the novel host[12]. Each of these processes becomes less likely if the two hosts differ more in terms of their geographic range, behaviour, and/or biochemistry (i.e., cellular receptors allowing viral attachment and invasion)[12,13]. Consequently, the probability that a pair of hosts will share a virus is shaped both by the species' underlying viral diversity and by species interactions represented by pairwise measures such as spatial overlap, phylogenetic relatedness, and ecological similarity[14–16].

Most previous investigations into pairwise determinants of viral sharing have been limited to one or two host orders (e.g., bats[17,18], primates[19], ungulates[16], and carnivores[14,16]), while sometimes lumping together different types of pathogens (e.g., helminths, viruses, and bacteria[14,16,19]). However, viruses can be shared across large host phylogenetic distances (e.g., Nipah virus in bats and pigs, among many others[20,21]), suggesting that a broad understanding of viral sharing across mammals is required to accurately predict patterns at different taxonomic and geographic scales. In addition, many mammalian orders have yet to be investigated in existing analyses—most notably rodents, which are highly diverse and host important zoonotic viruses[5,8]. In addition, although phylogenetic and geographic viral sharing effects have been empirically demonstrated, statistical models have not yet been used to validate viral sharing predictions using external datasets or make inferences about mammals with no known viral associations. If geographic and phylogenetic effects on viral sharing are as ubiquitous as they seem, these variables alone could provide a useful baseline model of viral sharing applicable across the mammal class.

Here, we analyse pairwise viral sharing using a novel, conservative modelling approach designed to partition the contribution of species-level traits from pairwise phylogeographic traits. This method of analysis stands in contrast to previous studies of mammalian viral sharing, which have mainly focussed on host-level traits, and importantly buffers against certain inherent biases in the observed viral sharing network, including host sampling bias, when making predictions.

## Results and discussion
**Predictors of viral sharing.** We fitted a model designed to partition the contribution of species-level effects and pairwise similarity measures to mammalian viral sharing probability. We used a published database of 1920 mammal–virus associations (excluding humans) as a training dataset[5]. These data included 591 wild mammal species, equalling 174,345 pairwise host species combinations, with 6.4% connectance—that is, 6.4% of species pairs shared at least one virus. We used a generalised additive

mixed model (GAMM) framework, including a species-level effect in our model as a multi-membership random effect, capturing variation in each species' connectedness and underlying viral diversity (see Methods). Overall, our model accounted for 44.8% of the total deviance in pairwise viral sharing, with 51.1% of this explained deviance attributable to the identities of the species involved (i.e., the species-level effect). Our model structure was effective at controlling for species-level variation in our dataset: when we simulated networks using just these parameters, species' predicted centralities were very close to their observed centrality (Supplementary Fig. 1). These results suggest that ~50% of the dyadic structure of observed viral sharing networks (in contrast to the true underlying network) is determined by uneven sampling and concentration on specific host species, and the remainder by macroecological processes.

As expected, increasing host phylogenetic similarity and geographic overlap were associated with increased probability of viral sharing across mammals, together accounting for the remaining 49% of explained model deviance (Fig. 1a–c). Geography, phylogeny, and their interaction all showed strong nonlinear effects, with geographic overlap in particular driving a rapid increase in viral sharing that began at ~0–5% range overlap values, peaked at 50% overlap values, and then levelled off (Fig. 1b). This effect closely mirrors previous observations of strong, nonlinear effects of geographic and phylogenetic similarity determining within-order viral sharing[14,16–19]. Curiously, we observed a downturn in sharing probability as closely related species exceeded 50% geographic overlap (Fig. 1b). However, this effect is of relatively limited importance in the context of our dataset: 93% of mammal pairs had less than 5% spatial overlap, while less than 0.5% had >50% overlap (Fig. 1b, d). The sparseness of data at this end of the distribution may also expose this effect to more unaccounted-for sampling biases, though some mechanistic explanations, such as apparent competition[22], are plausible. The great majority (86%) of mammal pairs in our dataset did not overlap geographically and rarely shared viruses unless phylogenetic similarity exceeded ~0.5 (Fig. 1a). This phylogenetic distance corresponds roughly to order-level similarity; that is, if two species did not overlap in space, it was highly unlikely that they shared a virus unless they were within the same taxonomic order (8% of pairs). Notably, phylogenetic similarity accounted for more than twice as much model deviance as did spatial overlap (33.8% vs 14.4%). The greater importance of phylogeny relative to geography contrasts with previous analyses concerning viral sharing in primates[19] and ungulates[16], likely reflecting the wider phylogenetic range of hosts considered here. This finding supports the important role of mammalian evolutionary history in shaping contemporary patterns of viral sharing and diversity[5,23].

In contrast to geography and phylogeny, minimum citation count and domestication status accounted for a vanishingly small amount of the deviance in viral sharing probability (0.2% and 0.1%, respectively) even though they have important effects on observed viral diversity in this dataset[5]. Their impacts on viral sharing may have been largely accounted for by the species-level random effects.

Our use of a pan-mammalian viral sharing dataset with a large sample size allowed us to investigate how geographic overlap and phylogenetic similarity affect viral sharing across different viral subgroups. These subgroups included RNA viruses, vector-borne RNA viruses, non-vector-borne RNA viruses, and DNA viruses. The importance of geographic overlap varied widely across all groups of viruses (Supplementary Fig. 2 and Supplementary Table 1), while the influence of host phylogenetic relatedness was more consistent (Supplementary Fig. 3 and Supplementary Table 1). Generally, host phylogeny was more important in

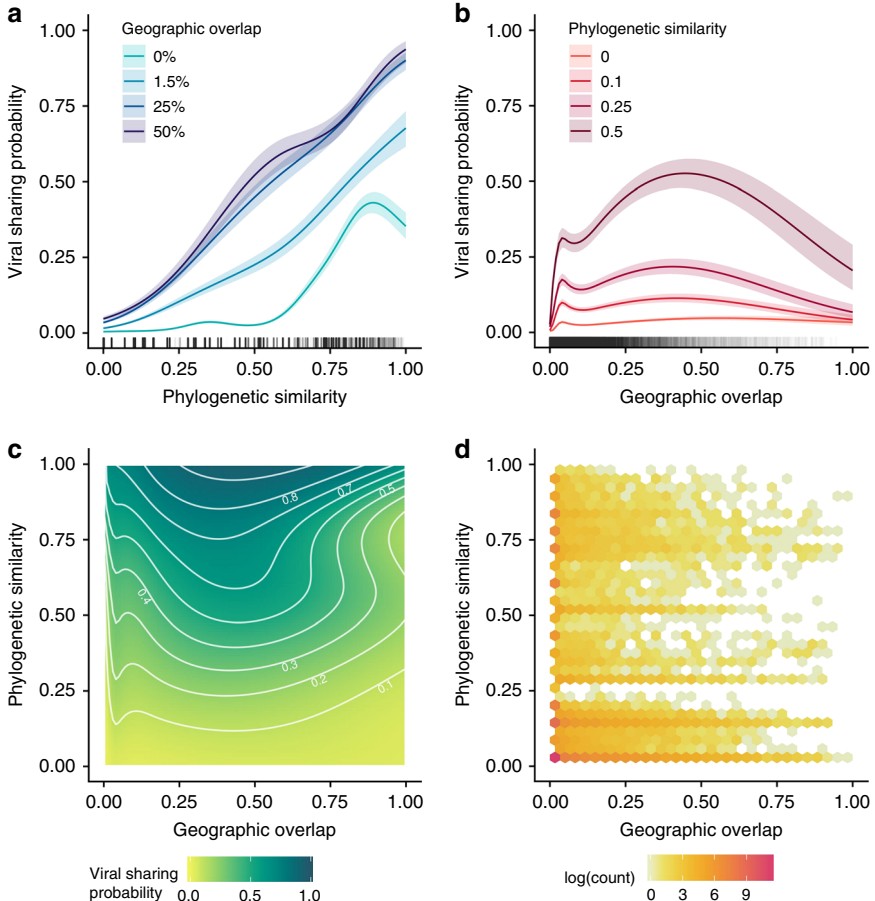

**Fig. 1 Viral sharing GAMM outputs and data distribution. a** Predicted viral sharing probability increases with increasing phylogenetic relatedness; the different coloured lines represent different geographic overlap values. **b** Predicted viral sharing probability increases with increasing geographic overlap; the different coloured lines represent different phylogenetic relatedness values. **c** The geographic overlap:phylogenetic similarity interaction surface, where the darker colours represent increased probability of viral sharing. White contour lines denote 10% increments of sharing probability. Labels have been removed from some contours to avoid overplotting. **d** Hexagonal bin chart displaying the data distribution, which was highly aggregated at low values of phylogenetic similarity and especially of geographic overlap.

determining sharing of DNA viruses than it was for RNA viruses, while space sharing was more important for vector-borne RNA viruses, and less so for non-vector-borne RNA viruses. These results likely reflect important aspects of viral ecology, transmission, and evolution. For example, RNA viruses are fast-evolving, allowing them to more quickly adapt to novel hosts, such that phylogenetic distances are less important in determining viral sharing patterns[24]. Conversely, DNA viruses are more evolutionarily constrained, with an evolutionary rate typically <1% that of RNA viruses[25], such that phylogenetic distance between hosts presents a more significant obstacle for sharing of DNA viruses. The profound importance of geographic overlap in shaping the viral sharing network for vector-borne RNA viruses (Supplementary Fig. 3) likely emerges from the geographic distributions and ecological constraints placed on vectors, lending further support to efforts to model the global spread of arboviruses by predicting changes in their vectors' distributions and ecological niches[26,27]. Generally, the fact that viral sharing across different viral subgroups was predicted by different macroecological relationships suggest they should be examined separately in future analyses where possible.

**Predicting pan-mammalian viral sharing.** Previous trait-based approaches used to model viral sharing and reservoir hosts have

been hindered by incomplete and inconsistent characterisation of traits central to those modelling efforts. In contrast, spatial distributions and phylogenetic data are readily available and uniformly quantified for the vast majority of mammals and, as we have shown, are reliable predictors of viral sharing (>20% of total deviance). Thus, we used our GAMM estimates to predict unobserved global viral sharing patterns across 8.8 million mammal–mammal pairs using a database of geographic distributions[28] and a recent mammalian supertree[29] (see Methods). The predicted network included 4196 (non-human) Eutherian mammals with available data, 591 of which were recorded with viral associations in our training data. We calculated each species' predicted degree centrality, as a simple and interpretable network-derived measure of viral sharing: that is, the number of other mammal species a given mammal species is expected to share at least one virus with. We identified geographic and taxonomic trends in degree centrality, validated our predicted sharing network using an external dataset, and simulated reservoir identification to assess host predictability for focal viruses (see Methods). For a visual representation of the predicted viral sharing matrix, see Supplementary Fig. 4.

We confirmed that our modelled network recapitulated expected patterns of viral sharing using the Enhanced Infectious Diseases Database (EID2) as an external dataset[30]. This dataset was constructed by mining web-based sequence data to identify

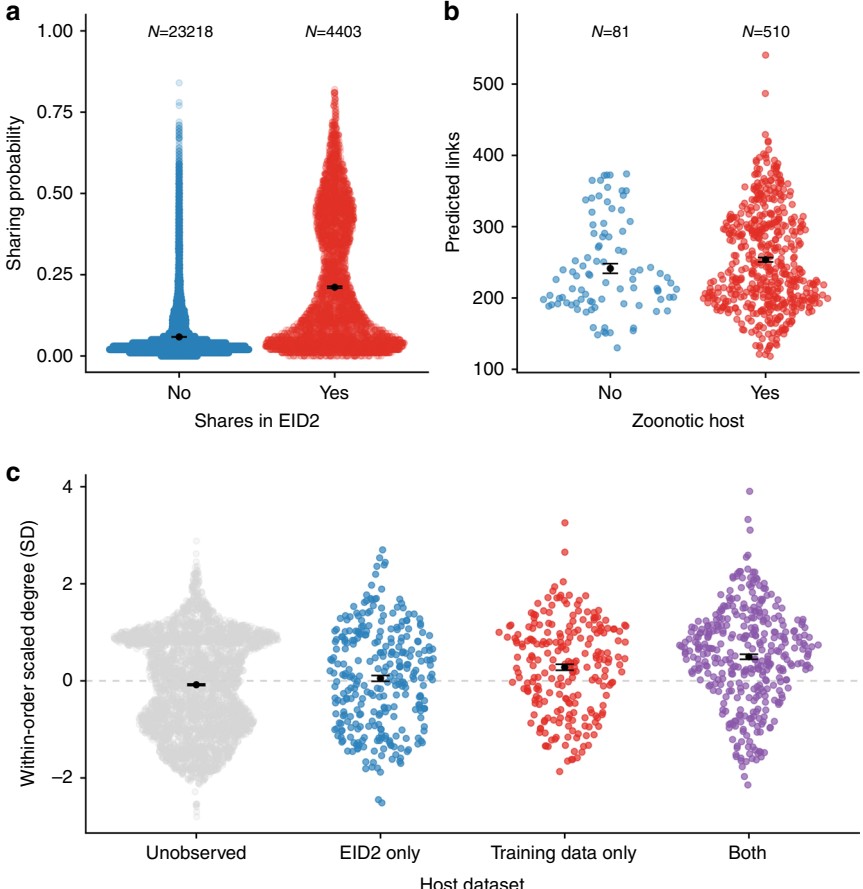

**Fig. 2 The predicted viral sharing network predicts observed trends in an independent dataset.** In all figures, points are jittered along the *x*-axis according to a density function; the black points and associated error bars are means ± standard errors. **a** Species pairs with higher predicted viral sharing probability from our model were more likely to be observed sharing a virus in the independent EID2 dataset. This comparison excludes species pairs that were also present in our training data. **b** Species that hosted a zoonotic virus in our dataset had more viral sharing links in the predicted all-mammal network than those without zoonotic viruses. **c** Species that had never been observed with a virus have fewer links in the predicted network than species that were known to host viruses in the EID2 dataset only, in our training data only, or in both. The *y*-axis represents viral sharing link number, scaled to have a mean of 0 and a standard deviation of 1 within each order for clarity. Black points represent means; error bars represent standard errors. Supplementary Figure 5 displays these same data without the within-order scaling.

host-pathogen associations, many of which are mammal–virus interactions[30]. Pairs of species that share viruses in EID2, but which were not in our training dataset (see Methods), had a much higher mean sharing probability in our predicted network (20% vs 5%; Fig. 2a). In addition, more central species in the predicted network were more likely to have been observed with a virus, whether zoonotic (Fig. 2b) or non-zoonotic (Fig. 2c), implying that the predicted network accurately captured realised the potential for viral sharing and zoonotic spillover. This finding concurs with similar work in primates that demonstrated that high centrality in primate–parasite networks is associated with carriage of zoonoses[31]. We corroborate these findings considering all mammal–mammal viral sharing links, not just zoonotic links, and show that for each mammalian order, species with higher degree centrality in our predicted network are more likely to have been observed with viruses in the EID2 dataset (Fig. 2c and Supplementary Fig. 5). Species with higher centrality in the global viral sharing network are likely more important for viral sharing overall, and thus have also been more likely to be observed with a (zoonotic) virus. Species that are more central in our predicted network could therefore be prioritised for zoonotic surveillance or sampling in the event of viral outbreaks with unknown mammalian origins. Given that mammal diversity predicts patterns of livestock disease[32] and zoonoses[33], the geographic

patterns of degree centrality predicted here (Fig. 3 and Supplementary Fig. 6; see below) could also be used as a coarse predictor of viral disease risk to livestock and human health, providing additional insights that emerge from the joint, nonlinear effects of geography and phylogeny as opposed to the examination of their effects in isolation. Similarly, where there is limited knowledge of mammalian host range for newly-discovered viruses, our modelled network can be used to prioritise the sampling of additional species for viral surveillance.

The high predicted centrality of known hosts may be due partly to selective sampling (i.e., viral researchers are more likely to sample common, wide-ranging host species that also share viruses with many other species[10,20]). This possibility is supported by the increased degree centrality for species that appear in both EID2 and our dataset rather than in only one of the two, as these species are presumably more well-known (Fig. 2c). Similarly, while we believe that our model was successful at accounting for variation in host-level diversity and study effort that influences network topology (see above; Supplementary Fig. 1), there are certain inherent biases in the training data which must be considered when interpreting our findings. Most notably, viral sharing estimates in our dataset may be affected by the fact that zoonotic discovery efforts commonly search limited geographic regions for a specific virus or group of viruses, artificially

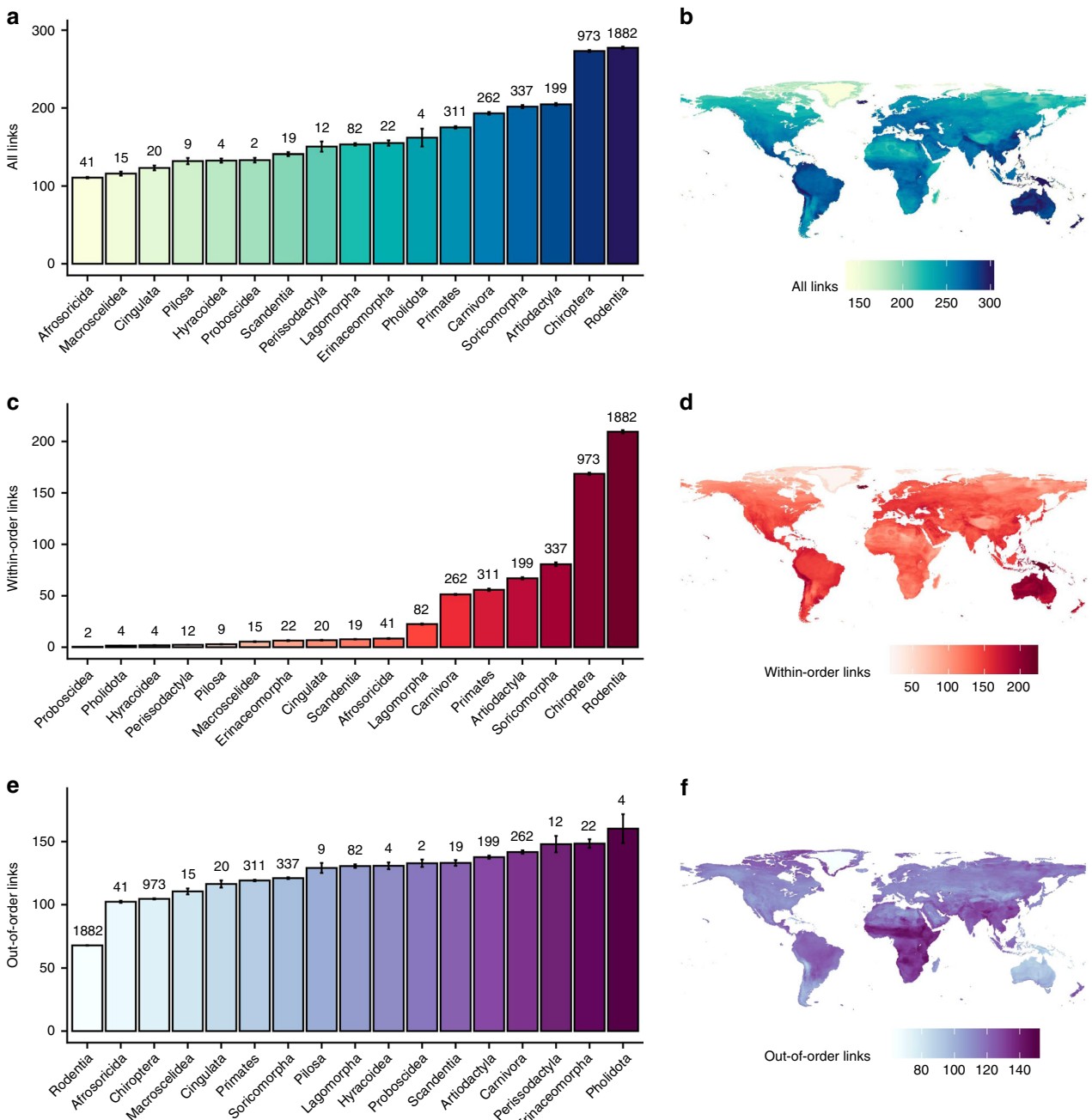

**Fig. 3 Taxonomic and geographic patterns of mean predicted viral sharing link numbers (degree centrality).** Top row: all viral sharing links; middle row: viral sharing links with species in the same order; bottom row: viral sharing links with species in another order. **a, c, e** Average species-level viral sharing link numbers for mammalian orders in our dataset. Bars represent means; error bars represent standard errors. **b, d, f** Geographic distributions of mean viral sharing link numbers. Distributions were derived by summing the viral sharing link numbers of all species inhabiting a 25 km² grid square and dividing them by the number of species inhabiting the grid square, giving mean degree number at the grid level.

increasing the likelihood of detecting these viruses in the same region compared to a geographically random sampling regime. Moreover, when a mammal species (e.g., a bat) is found with a focal virus (e.g., an ebolavirus), it is logical for researchers to then investigate similar, closely related species in nearby locales[34]. These sampling approaches could disproportionately weight the network towards finding phylogeographic effects on viral sharing probability. However, it is highly encouraging that our model predicted patterns in the external EID2 dataset, which was constructed using different data compilation methods but also comprises global data covering several decades of research[30]. In sum, we believe that our approach is a conservative method for

minimising the biases inherent in the data. The knowledge that the observed mammalian virome is biased ultimately calls for more uniform viral sampling across the mammal class and increased coverage of rarely-sampled groups, lending support to ongoing efforts to systematically catalogue mammalian viral diversity[3].

**Taxonomic and geographic patterns of predicted viral sharing.** Our network predicted strong taxonomic patterns in the probability of viral sharing. Looking across mammalian orders, rodents (Rodentia) and bats (Chiroptera) had the most predicted species-level viral links, while carnivores and artiodactyl

ungulates had substantially fewer (Fig. 3a). Examining multiple mammalian orders allowed us to partition the predicted sharing network into within- and between-order links to investigate whether certain orders are better-connected to other orders. Indeed, this partitioning revealed differences in taxonomic and geographic patterns of viral sharing. In bats and rodents, large numbers of within-order links are driven by high within-order species diversity (Fig. 3c). Interestingly, when within-order links were ignored, leaving only out-of-order links, rodents and bats were among the least-connected Eutherian orders (Fig. 3e), while even-toed ungulates and carnivores were ranked among the most-connected (Fig. 3e). Taken together, these results imply that while bats and rodents are important in viral sharing networks, their sharing is mainly restricted to other bats and rodents, respectively. This distinction only applied to mean link numbers; when link numbers were summed, rodents and bats remained highly connected regardless of which metric was used, as a result of their species richness (Supplementary Fig. 6).

Previous analyses have demonstrated that both bats and rodents are important for hosting zoonotic viruses, with possible explanations including species-level phenotypic traits such as behaviour[5], life history[9], or metabolic idiosyncracies[35]. Our results imply that while both orders potentially host many zoonoses purely as a result of their species richness (Supplementary Fig. 6), the vast majority of their viral sharing occurs within-order even though larger phylogenetic jumps are necessary for human spillover. Intriguingly, recent work has shown that infection of an aggregated phylogenetic selection of hosts is an important contributor to viral zoonotic potential[36]. Rodents' and bats' tendency towards high viral interconnectedness could encourage viruses to achieve such aggregation, leading to opportunities for spillover into humans. In our analysis, both orders' high centrality emerged purely as a result of their phylogenetic diversity and geographic distributions, rather than from other phenotypic traits. If well-connected species in our network are more likely to maintain a high diversity of viruses (e.g., via multi-host dynamics that help to maintain a threshold population size[37]), this may contribute to the high viral diversity documented in bats and rodents[5]. Efforts to prioritise viral sampling regimes should consider biogeography and mammal–mammal interactions in addition to searching for species-level traits associated with high viral diversity.

Encouragingly, our network showed predictable scaling laws similar to those of other known ecological networks[38]. Viral link numbers in within-order subnetworks (e.g., between different bat species) correlated strongly with species diversity within each order ($R^2 = \sim 0.85$), following a power law with a $Z$ value of ~0.8 (Supplementary Fig. 7). Similarly, out-of-order links (e.g., between a bat and a rodent) scaled linearly with the product of the species richness of both orders (Supplementary Fig. 8).

To visualise geographic patterns of viral sharing, we projected species-level degree centrality across the species' ranges, then calculated grid cell-level mean degree centrality (Fig. 3b), as well as summed degree centrality (Supplementary Fig. 6). Average centrality peaked in tropical areas of South and Central America, Sub-Saharan Africa, and Southeast Asia, especially in the Andes and Himalayas (Fig. 3b). These patterns align with previously-reported hotspots of emerging zoonoses and predicted viral diversity[5,33] and imply that areas of high biodiversity are centres of viral sharing not just because of the number of overlapping species (i.e., high species richness), but also because more closely related species create a more connected viral sharing network in these areas. This densely-connected network structure and the increased biomass present in the tropics might have synergistic implications for cross-species maintenance and transmission of viral diversity in these areas. The geographic distributions of mean predicted within- and between-order viral links differed notably from the distribution of interspecific links generally: the

relative importance of South America and East Asia was higher for within-order links (Fig. 3d), while Sub-Saharan Africa remained a hotspot for out-of-order links (Fig. 3f). Geographic patterns of summed link numbers more closely mirrored underlying host species richness, whether for all links, within-order links, or out-of-order links (Supplementary Fig. 6).

We acknowledge that our phylogeographic model of viral sharing does not account for complex ecological interactions such as coinfection or coevolution, which could impact how patterns of exposure and host susceptibility translate to realised viral diversity. Future investigations could extend our framework to simulate the dynamic co-speciation of mammals and their viruses in order to account for these processes and/or to explicitly investigate how viral sharing connectivity and viral diversity are correlated across mammal species. Our model may also prove useful for building and parameterising much-needed multi-host network models for conservation purposes, particularly where there is scarce prior information on interspecific pathogen sharing[37,39].

**The network as a predictive tool**. Identifying potential hosts for known and novel viruses is an important component of pre-emptive zoonotic disease surveillance that can speed public health responses. Predictive techniques based on species-level phenotypic and genomic data have been suggested to help prioritise sampling targets[6,7,9]. To augment these approaches representing a promising methodological advance, the network approach captures the additional mechanistic and ecological underpinnings of viral sharing. We therefore interrogated our predicted viral sharing network to investigate whether it could be used to identify potential hosts of known viruses at the species level.

We investigated the predictive potential of our model by iteratively selecting all but one of the known hosts for a given virus, then using the predicted sharing patterns of the remaining hosts to identify how the focal (removed) host was ranked in terms of its sharing probability (see Methods). Our model showed a surprisingly strong ability to predict observed host species for 250 viruses with at least two known (non-human) mammal hosts. In practical terms, these species-level rankings could set sampling priorities for public health efforts seeking to identify hosts of a novel zoonotic virus, where one or more hosts are already known. Across all 250 viruses, the median ranking of the left-out host was 72 out of a potential 4196 mammals (i.e., in the top 1.7% of potential hosts). To compare this ranking to alternative heuristics, we examined how high the focal host would be ranked using phylogenetic relatedness or spatial overlap values alone (i.e., the most closely-related, followed by the second-most-related, etc.). Using this method, the focal host was ranked, on average, 288th (for phylogeny) or 283rd (for space), identifying the focal host in the top 7% of potential hosts and demonstrating that sampling prioritisation schemes based on our phylogeographic model would require only 1/4 as many sampling targets in order to identify the correct sharing host. Our model therefore represents a substantial improvement over search methods based only on spatial or phylogenetic similarity. Our model performed similarly at identifying focal hosts in the EID2 dataset[30]: for the 109 viruses in the EID2 dataset with more than one host, the focal host was identified in the top 63 (1.5%) potential hosts. In contrast, ranked spatial overlap predicted the focal host in the top 560 hosts, and phylogenetic relatedness in the top 174.

We observed substantial variation in our model's ability to predict known hosts among different viruses. For example, the correct host was predicted first in every iteration for seven viruses and in the top 10 hosts for 42 viruses. Results for 128 viruses had the focal host falling within the top 100 guesses, and for only six viruses were the model-based host searches worse than chance

(i.e., the focal host ranked lower than 50% of all mammals in terms of sharing probability). We used this measure of viral sharing "predictability" to investigate whether certain viral traits affected the ease with which phylogeography predicted their hosts. Viruses with broad host phylogenetic ranges challenge reservoir prediction efforts since many more species must often be sampled before identifying the correct host(s). To investigate whether the predictive strength of our model was limited for viruses with broad host ranges and/or other viral traits, we fitted a linear mixed model (LMM) which showed a strong negative association between viruses' known phylogenetic host breadth and the predictability of focal hosts (model $R^2 = 0.70$; host breadth $R^2 = 0.67$; Supplementary Fig. 10). This association demonstrates, unsurprisingly, that predicting the hosts of generalist viruses is intrinsically difficult. This adds a potential limitation to the applicability of our network approach, given that zoonotic viruses commonly exhibit wide host ranges[2,5]. A family-level random effect accounted for little of the apparent variance in predictability among viral families (Supplementary Fig. 9).

Once viral host range was accounted for, hosts of vector-borne viruses were slightly easier to predict than non-vector-borne viruses ($R^2 = 0.1$; Supplementary Fig. 10)—perhaps because the sharing of vector-borne viruses depends more heavily on host geographic distributions (Supplementary Fig. 3). Despite additional variation in the data, no other viral traits (e.g., RNA vs DNA, segmented vs non-segmented) were important in the LMM. This implies that host phylogeographic traits are a good broad-scale indicator of viral sharing, particularly when ecological specifics of the virus itself are unknown.

**Conclusions**. In summary, we present a simple, highly interpretable model that predicted a substantial proportion of viral sharing across mammals and is capable of identifying species-level sampling priorities for viral surveillance and discovery. It is worth noting that the analytical framework and validation we describe were conducted on a global scale, while many zoonotic sampling efforts occur on a national or regional scale. Restricting the focal mammals to a regional pool may alter the applicability of our model in certain sampling contexts, and future studies could leverage higher-resolution phylogenetic and geographic data to fine-tune predictions. In particular, the mammalian supertree[29] has relatively poor resolution at the species tips such that relatedness estimates based on alternative molecular evidence (e.g., full genome data from hosts) may allow more precise estimates of the phylogenetic relatedness effect on viral sharing. Alternatively, our model could be augmented with additional host, virus, and pairwise traits, using similar pairwise formulations of viral sharing as a response variable. Such model augmentations may better identify ecological specificities that are critical for the transmission of certain viruses, allow for partitions by viral subtypes, and, ultimately, may increase the accuracy of host predictions. By generalising the spatial and phylogenetic processes that drive viral sharing, our model serves as a useful guide for the prioritisation of viral sampling, presenting a baseline for future modelling efforts to compare against and improve upon.

Our ability to model and predict macroecological patterns of viral sharing is important in an era of rapid global change. Under all conceivable global change scenarios, many mammals will shift their geographic ranges, whether of their own volition or through human assistance. Mammalian parasite communities will likely undergo considerable rearrangement as a result, with potentially far-reaching ecological and health consequences[40–43]. Our findings suggest that novel species encounters will provide opportunities for interspecific viral transmission, which could be facilitated by even relatively small changes in range overlap. These future cross-species transmission events will have profound implications for conservation and public health, potentially devastating populations of host species without evolved resistance to novel viruses (e.g., red squirrel declines brought about by parapoxvirus infections spread by introduced grey squirrels[22]) or increasing zoonotic disease risk by introducing viruses to human-adjacent amplifier hosts (e.g., horses increasing the risk of human infection with Hendra virus[20]). Thus, our global model of mammalian viral sharing provides a crucial complement to ongoing work modelling the spread of hosts, vectors, and their associated diseases as the result of climate change-induced range expansions[26,40,43].

## Methods

**Making the training data network**. Our dataset included 1920 mammal–virus associations obtained from an exhaustive literature search which has been used to investigate how species traits influence mammalian viral diversity[5]. We removed humans and rabies virus from the dataset as both were disproportionately well-connected, and we removed 20 non-Eutherian mammals because they were extreme phylogenetic outliers, leaving 591 Eutherian mammals that shared 401 viruses. We made an unweighted bipartite network using the mammal–virus associations and projected the unipartite mammal–mammal network, which we then converted into a sequence of all unique mammal–mammal pairs where 1/0 denoted whether the pair of species shared a virus or not. This comprised only the lower triangle of the adjacency matrix to avoid duplicating associations and to remove self-connections, and only included mammals with at least one sharing link (final $N = 174,345$ unique mammal–mammal pairs). 6.4% of these pairs shared at least one virus.

All analyses were performed in R version 3.6.0[44]. Phylogenetic similarity was calculated using a mammalian supertree[29]. Pairwise phylogenetic distances were defined as the cumulative patristic distance between the two species and were scaled to between 0 and 1, and subtracted from 1 to give a measure of relative phylogenetic similarity (rather than distance). Of the 4716 Eutherian species in the mammalian supertree, 591 had virus association records in our fully-connected network and 4196 had known geographic ranges. We used IUCN species ranges to quantify species' geographic distributions[28]. These range maps are generated based on expert knowledge and only comprise species presence/absence information rather than density. We converted all range polygons to 25 km$^2$ raster grids. For each species-pair, we quantified range overlap as the number of raster grid squares jointly inhabited by the two species (in the Mollweide projection, which exhibits equal grid size), divided by the total number of grid squares occupied by these species combined, so that each value was scaled from 0 to 1: $\text{overlap}_{A,B} = \text{grid}_{A,B}/(\text{grid}_A + \text{grid}_B - \text{grid}_{A,B})$. Disease-related research effort for each host species was quantified using counts of studies including species names and disease-related terms such as "virus", "pathogen", or "parasite"[5]. To fit citation number as a pairwise trait, we took the smaller of a pair of species' respective citations, and log-transformed the value. Domestication status was defined *sensu lato*, based on whether a species was ever seen in a domestic setting[5]. We fitted this as a binary pairwise trait where 1 = at least one of the species was domesticated and 0 = neither species had been domesticated.

**Model formulation**. We fitted a Generalised Additive Mixed Model (GAMM) to examine which traits influenced viral sharing among mammal pairs using accelerated discretized implementation in the **mgcv** package[45]. We fitted viral sharing (0/1) as the response variable, with a binomial family specification. The model had the following structure:

$$\text{Bernoulli(viral sharing)} \sim s(\text{phylogenetic similarity}, \text{by} = \text{ordered}(Gz))$$
$$+ \ t2(\text{phylogenetic similarity, geographic overlap}, \text{by} = \text{ordered}(!Gz))$$
$$+ \ \text{minimum citation number} + \text{domestication status}$$
$$+ \ mm(\text{species 1} + \text{species 2})$$

The first term ("s") represents a phylogeny effect smooth fitted across species pairs that did not overlap in space ($Gz = 1$), and "t2" represents a phylogeny: geography tensor product smooth fitted to species that had geographic overlap greater than zero ($Gz = 0$). This allowed us to model these two data partitions separately, helping us to more effectively model a large number of spatial zeroes (85% of species pairs did not overlap in space). "mm" represents a multi-membership random effect, accounting for the identity of both species in the pair. The effects are similar to a categorical random effect, but each data point can have multiple categorical levels for the effect, which are constrained to sampling from the same limited distribution. We implemented this multi-membership effect to control for species-level effects by including a species-level effect for both the row (species 1) and column (species 2) of the sharing matrix. Using the paraPen specification in **mgcv**, these random effects were constrained to sample from the same distribution, resulting in a single estimate of the variance associated with each unique species despite some species being coded many more times as species 1 than

as species 2, and vice versa. Multi-membership random effects have been used in interaction network contexts before to control for row and column effects—for example, in a study of disease transmission in a chimpanzee social network[46]. Most precisely, these effects in our model help capture variation in viral sharing that could likely be explained by species-level factors that are unobserved or otherwise excluded (i.e., differences in underlying viral diversity, which would be expected to positively impact the probability of interspecific sharing). In sum, this model formulation allowed us to estimate the effect of pairwise predictors (geographic overlap, phylogenetic similarity) in determining viral sharing as well as evaluate the influence of species identity. Although there are many ways of modelling dyadic traits such as viral sharing, we chose to use this approach because of its relative simplicity, similarity to the linear and additive models used in previous studies[14–16,18,19,47], and its ability to untangle dyadic and pairwise contributions to viral sharing in a highly interpretable manner.

We elected to use a binary model of viral sharing (0/1) rather than an integer count model (0+) for two reasons. First, the data distribution was highly skewed, with few very large values and many zeroes. Under these conditions, we found a count-based model formulation including species-level random effects computationally intractable. Second, the observed viral diversity is likely a considerable underestimate, but the extent of this underestimate is a matter of hot debate[3,4]. As such, the predictions for viral sharing from such a model could be highly relative and biased, while binary models offer a more appropriate resolution to quantify sharing patterns. We therefore avoided estimating a precise number of viruses shared among pairs of species.

To investigate whether the effects of geography and phylogeny depended on which subset of viruses we investigated, we fit the model to non-exclusive subnetworks of mammal–mammal pairs based on the types of viruses they were connected by. Viral subtypes included RNA viruses (566 hosts sharing 381 viruses); vector-borne RNA viruses (333 hosts sharing 164 viruses); non-vector-borne RNA viruses (391 hosts sharing 205 viruses); and DNA viruses (151 hosts sharing 205 viruses). There were only two vector-borne DNA viruses in our data. We eliminated from each analysis any hosts that were not carrying the focal virus type.

**Model validation**. To check the fit of the model, we predicted 0/1 viral sharing values from the model 1000 times and examined how the values compared to the proportions of 0's and 1's in the observed data, finding high agreement between the two. We repeated this procedure using (a) the full dataset; (b) only the fixed effects, with random effects randomised in each iteration; and (c) only the random effects, with fixed effects held at the mean. We then used these predicted links to create 1000 unipartite viral sharing networks, estimating link numbers (degree centrality) for species in each replicated network. We took the mean of these values across the 1000 replicated networks to give the predicted values displayed in Supplementary Fig. 1.

We quantified deviance contributions of our explanatory variables by calculating model deviance when dropping each variable, and comparing these against deviance values for the full model and an intercept-only model. For each of our explanatory variables (geographic overlap, phylogenetic similarity, minimum citation number, domestication status, and species-level random effects) we randomised the observed values 1000 times, then predicted sharing probabilities for these values using our model estimates. This randomisation procedure allowed us to predict while accounting for the uneven data distribution, rather than using mean values.

**Simulating viral sharing networks**. Following reconstruction of the observed network as part of our model validation, we repeated the prediction process on an exhaustive mammal dataset to estimate viral sharing across all mammals. We set minimum citation number to the data mean, and set domestication status to 0. We repeated the predictions 1000 times, randomising the species-level random effects each time. The full prediction dataset included 4196 Eutherian mammals with known spatial distributions and phylogenetic associations, resulting in 8.8 million unique pairwise combinations. After predicting 1000 binary sharing networks across all mammals, we summarised the average predicted link number (degree centrality) of each species across the 1000 replicates. We then calculated the mean species-level link number within each mammalian order to examine taxonomic patterns. To project the spatial patterns of connectedness, we assigned each species range polygon the link number (degree centrality) of its host species[28] and took the mean value for each grid square, thereby correcting for species richness. We then repeated these taxonomic and geographic summaries using within-order and between-order link numbers separately. We also took the summed values, which more closely reflect underlying patterns of species richness.

We validated the predicted network by comparing it to sharing patterns in the Enhanced Infectious Diseases Database (EID2)[30]. We eliminated species pairs that were in our training data and identified whether species pairs that shared viruses in EID2 were more likely to share viruses in our predicted network than species pairs that did not. In addition, we investigated whether species that were shown to host zoonoses in our training dataset were more highly-connected in the predicted network. Finally, we investigated whether species that were present in only EID2, in only our training data, or in both were more highly-connected in our predicted network than species that did not appear in either dataset and were therefore taken to have not been observed hosting a virus.

**Predicting hosts of focal viruses**. To investigate the ability of the model to predict known hosts of viruses in our dataset, we iteratively investigated the sharing patterns of known hosts independently for all viruses with >1 host. For each virus, we removed one host at a time, and then investigated which species the remaining known host species were likely to share viruses based on the all-mammal predicted network. If the removed host ("focal host") was on average highly likely to share viruses with the remaining species, our model was taken to be useful for predicting patterns of mammalian viral sharing. The mean ranking of the focal hosts across each prediction iteration was used as a measure of "predictability" for each virus. We carried out this process for the 250 viruses with more than one known host with associated geographic and phylogenetic data and then on the 109 such viruses in the EID2 data.

Once the predictability of each virus was calculated, we fitted a linear mixed model examining log10(mean focal host rank) as an inverse measure of predictability (higher rank corresponds to decreased predictability) for each virus. We added mean phylogenetic host similarity as a fixed effect and viral family as a random effect to quantify how viral phylogeny affected predictability. We included additional viral traits in the model, including cytoplasmic replication (0/1); segmentation (0/1); vector-borne transmission (0/1); double- or single-strandedness; DNA or RNA; enveloped or non-enveloped; and zoonotic ability (0/1 for whether the virus was associated with humans in our dataset).

**Reporting summary**. Further information on research design is available in the Nature Research Reporting Summary linked to this article.

## Data availability

Data for all analyses are available at https://doi.org/10.5281/zenodo.3745672.

## Code availability

Code for all analyses is available at https://doi.org/10.5281/zenodo.3745672.

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

## Acknowledgements

This work was conducted during a placement funded by the National Environmental Research Council (NERC) Overseas Research Fund awarded to G.F.A. G.F.A.'s PhD studentship was likewise funded by NERC (Grant Number: NE/L002558/1). E.A.E., N.R., and K.J.O. were funded by the generous support of the American people through the United States Agency for International Development (USAID) Emerging Pandemic Threats PREDICT project. Additional support was provided by the National Institute of Allergy and Infectious Diseases of the National Institutes of Health (Award Number R01AI110964) and the US Department of Defense, Defense Threat Reduction Agency (HDTRA11710064). The authors thank Colin Carlson, Verity Hill, and members of EcoHealth Alliance for advice and helpful comments on the manuscript.

## Author contributions

G.F.A., E.A.E., N.R., and K.J.O. designed the study together. G.F.A. conducted the analyses under the supervision of N.R., E.A.E., and K.J.O. G.F.A. wrote the manuscript, while E.A.E., N.R., and K.J.O. offered comments and edits to the manuscript throughout.

## Competing interests

The authors declare no competing interests.
