## [Peer Review File · Nature Communications]

Editorial Note: This manuscript has been previously reviewed at another journal that is not operating a transparent peer review scheme. This document only contains reviewer comments and rebuttal letters for versions considered at Nature Communications .

Reviewers' Comments:

Reviewer #1:

Remarks to the Author:

My only previous concern was that the context of this study in the wider body of existing work was poorly characterized. This has been addressed, and I have no further concerns.

Reviewer #4:

Remarks to the Author:

This manuscript aims to examine the phylogenetic and host determinants of viral sharing between mammals. This is an interesting and well-written manuscript that should appeal to the readership of Nature Communications. The results aren't particularly surprising or novel in respect to what is known about parasite sharing, but I think that the approach is quite interesting (whilst I do have a few concerns below) and dataset robust. In particular, the authors should be congratulated on testing their predictions using a separate dataset - this is rarely done. In general, the prediction part of this manuscript is really well done. I do have a few concerns, however on other parts of this ms. Mainly, the use of GAMMs on essentially a collection of distance matrices seems far from optimal. Given that the 'species effect' was so strong (51% of the explained deviance), using an approach that accounts naturally for these dependencies, such as a Markov Random Fields or Generalized Dissimilarity Models (or a myriad of related approaches) seems warranted (see Clark and Clegg, 2017). Then the viral count data could be included and the resultant matrix could be turned into (dis)similarity matrix and transformed appropriately to account for the skew the authors mention. Losing the count data seems like an avoidable loss. At a minimum, I think more explanation is needed regarding multi-membership random effects - I had to do some research to find out how they worked. As they are quite rarely used in ecology, I think a sentence or two to give a brief overview would help.

Perhaps the biggest suggestion is that the authors use 'viral sharing network' throughout the text but we never see the network. Visualizing the network - on even parts of it - is not only a useful sanity check but also crucial for looking at how the network is structured. Nodes could be coloured by phylogenetic group with the size determined by the number of shared viruses divided by citations (or some such scheme). As this network is large, spectral decomposition of the Laplacian matrix may help visualize the structure. Even if the figure isn't in the main text, I feel like it would be a useful addition to the supps.

Other minor comments:

Fig 1B: Why do you think viral sharing probability decline so markedly with phylogenetically similar species (0.5) which have high overlap? Seems counterintuitive. Also why did you choose 0, 0.1, 0.25, 0.5? Seems like high values would be more interesting. Same with Fig. 1A.

48: Fountain-Jones et al 2019 as well for carnivores.

50: Citations?

80: Strong impact on the centrality? I feel like this sentence needs a bit more explanation here.

136: Citations needed.

152: Ref 28 is not a super-tree paper (it is the Fritz et al 2009 paper). Which super-tree did you use?
Suggest checking other citations.

534: Patristic distance?

Best,

Nick Fountain-Jones

Reviewers' comments:

Reviewer #1 (Remarks to the Author):

My only previous concern was that the context of this study in the wider body of existing work was poorly characterized. This has been addressed, and I have no further concerns.

- We thank Reviewer #1 for their previous critiques and for reviewing our revised ms.

Reviewer #4 (Remarks to the Author):

This manuscript aims to examine the phylogenetic and host determinants of viral sharing between mammals. This is an interesting and well-written manuscript that should appeal to the readership of Nature Communications. The results aren't particularly surprising or novel in respect to what is known about parasite sharing, but I think that the approach is quite interesting (whilst I do have a few concerns below) and dataset robust. In particular, the authors should be congratulated on testing their predictions using a separate dataset - this is rarely done. In general, the prediction part of this manuscript is really well done. I do have a few concerns, however on other parts of this ms. Mainly, the use of GAMMs on essentially a collection of distance matrices seems far from optimal. Given that the 'species effect' was so strong (51% of the explained deviance), using an approach that accounts naturally for these dependencies, such as a Markov Random Fields or Generalized Dissimilarity Models (or a myriad of related approaches) seems warranted (see Clark and Clegg, 2017). Then the viral count data could be included and the resultant matrix could be turned into (dis)similarity matrix and transformed appropriately to account for the skew the authors mention. Losing the count data seems like an avoidable loss. At a minimum, I think more explanation is needed regarding multi-membership random effects – I had to do some research to find out how they worked. As they are quite rarely used in ecology, I think a sentence or two to give a brief overview would help.

- We greatly appreciate Reviewer #4's (Dr Fountain-Jones) expertise and familiarity with Generalised Dissimilarity Models (GDMs) and other methods. We have carefully considered his suggestion to use alternative modelling approaches to improve our mammalian viral sharing network. Dr Fountain-Jones' main reason for suggesting GDMs is the ability to control for sampling bias. He additionally points out that using count data for viral sharing events would provide more information. We indeed originally included GDM's in our list of potential approaches when we first considered the question and began this research. While we realise that our approach is not especially common – although not unprecedented – in network studies, we have opted to retain our use of GAMMs and binary data for a variety of reasons that we have taken care to outline in detail below.

- First, and most importantly, we chose to model the viral sharing network as a binary (rather than weighted, or count-based) network for empirical reasons. We outlined those reasons at lines 617-625 and in our previous response to the reviewers at *Nature Ecology and Evolution*. Namely, the observed viral sharing network is prohibitively poorly-sampled, and so we aimed to avoid prescribing or predicting the number of viruses shared, particularly because this may influence priorities for viral discovery. As such, the probability of viral sharing alone is a more parsimonious option for the analytical aims of our study. Further, much of the variation in number of known viruses would be highly heterogeneous across species, exacerbating the unevenness of sampling in the data. As such, the choice to model the network as a binary was not a necessity of our general modelling approach, but resulted from consideration of the mechanisms generating variation in viral counts across host species. For these reasons, among others, all references that we cite below (Davies & Pedersen 2008; Huang *et al.* 2014; Willoughby *et al.* 2017; Wells *et al.* 2018, 2020; Stephens *et al.* 2019) use either binary viral sharing or proportion of shared viruses as a response variable, rather than viral counts.
- We also have several methodological reasons for choosing GAMMs with multi-membership random effects as our approach of choice. First, GAMMs are more similar to the methods already used for parasite sharing models across and within mammal orders. Our study expands upon previous parasite sharing analyses (Davies & Pedersen 2008; Huang *et al.* 2014; Willoughby *et al.* 2017; Wells *et al.* 2018, 2020; Stephens *et al.* 2019); their statistical methods included: a generalised linear model (GLM) or similar model with no specialised random effects (Davies & Pedersen 2008; Huang *et al.* 2014; Willoughby *et al.* 2017; Wells *et al.* 2018), a generalised additive model (GAM) with no specialised random effects (Stephens *et al.* 2019), and a hierarchical regression with an order-level random effect (Wells *et al.* 2020). However, none of these other studies attempted to use more sophisticated models than GLMs/GAMs. Multi-membership random effects are an established method for controlling for row and column effects in networks analysed using GLM frameworks: for example, (Rushmore *et al.* 2013) used multi-membership random effects to control for individual-level non-directionality in networks of chimpanzee interactions. As such, our model is functionally similar to prior models but is more conservative and represents a useful extension of the pre-existing viral sharing analysis methodologies. We believe our approach is analytically robust, well-founded, simple to expand upon, and easy to connect to previous analyses, all while tangibly improving on these previous approaches.
- Finally, we used this approach because it allows for easy comparison of the contribution of species-level effects vs. pairwise host traits to viral sharing,

which was a central focus of our study and something we aimed to quantify (deviance explained) rather than merely control for. Further, we were then able to simulate a viral sharing network explicitly avoiding this effect. The species-level effect indeed accounted for a large proportion of the deviance explained, as Dr Fountain-Jones points out, but the effect also left a considerable amount of deviance explained by the fixed effects. This is the portion of the model that we used to simulate with, explicitly avoiding the species-level variation, and it would be difficult to do so with another approach, particularly if using count or dissimilarity data. In addition, research effort itself surprisingly had no detectable effect on the probability of sharing, despite the importance of the species-level effect, and we find it unlikely that GDMs would make this easier to interpret, more reliable, or more accurate.

- Fundamentally, there are many valid ways to approach this problem statistically, all of which result in some tradeoff between accuracy, interpretability, and simplicity. We believe that our approach represents an appropriate balance of these concerns with respect to our scientific question. Other approaches, such as GDMs, but also Exponential Random Graph Models (ERGMs) or Latent Space Models (LSMs), doubtless will produce additional insights but we respectfully do not believe they would have yielded more than slight differences from our results here.
- As suggested by Dr. Fountain-Jones, we have added several sentences to the methods (lines 590-615) to outline further the basis of multi-membership effects, the other choices of models, and the reasons for selecting GAMMs for our approach.

Perhaps the biggest suggestion is that the authors use ‘viral sharing network’ throughout the text but we never see the network. Visualizing the network – on even parts of it – is not only a useful sanity check but also crucial for looking at how the network is structured. Nodes could be coloured by phylogenetic group with the size determined by the number of shared viruses divided by citations (or some such scheme). As this network is large, spectral decomposition of the Laplacian matrix may help visualize the structure. Even if the figure isn’t in the main text, I feel like it would be a useful addition to the supps.

- We appreciate that a depiction of the viral sharing network will help readers to visualise the processes at play, and so we have included a new supplementary figure of the sharing matrix, coloured according to dyadic sharing probability (now Figure SI4).

Other minor comments:

- Fig 1B: Why do you think viral sharing probability decline so markedly with phylogenetically similar species (0.5) which have high overlap? Seems counterintuitive.
 - We originally had included text in our manuscript to interpret the downturn in viral sharing with higher spatial overlap values; however, we removed it based on previous reviewers' comments. We have added some of this interpretation back in, lines 107-120, while also pointing out that the data in this area represents only a very small proportion of the dataset as a whole:
 - “Curiously, we observed a downturn in sharing probability as closely related species exceeded 50% geographic overlap (Figure 1B). However, this effect is of relatively limited importance in the context of our dataset: 93% of mammal pairs had less than 5% spatial overlap, while less than 0.5% had >50% overlap (Figure 1B,D). The sparseness of data at this end of the distribution may also expose this effect to more unaccounted-for sampling biases, though some mechanistic explanations, such as apparent competition²², are plausible.”
- Also why did you choose 0, 0.1, 0.25, 0.5? Seems like high values would be more interesting. Same with Fig. 1A.
 - We chose these values because they provide the most interesting cuts across the 3D surface displayed in panel C, weighed against the data distribution displayed in panel D. Very few species have very high phylogenetic similarity or especially high geographic overlap, so that we decided not to display trends for these very high values, as they were relevant to very little of the data. The entire probability surface is available in panel C for those that are interested.
- 48: Fountain-Jones et al 2019 as well for carnivores.
 - We appreciate this suggested reference and assume this refers to “Endemic infection can shape exposure to novel pathogens: Pathogen co-occurrence networks in the Serengeti lions”?
 - If so, this sentence is referring to within-order, interspecific viral sharing, in contrast to the suggested Fountain-Jones reference, which examines within-species, between-individual sharing of parasites. As such, this reference does not fit with the theme of the sentence, which we have left unaltered.
- 50: Citations?
 - We have added citations to clarify which subset we are referring to here.
- 80: Strong impact on the centrality? I feel like this sentence needs a bit more explanation here.
 - We have rephrased this sentence to clarify:
 - “Our model structure was effective at controlling for species-level variation in our dataset: i.e., when we simulated networks using just

these parameters, species' centrality in the networks were extremely close to their observed centrality (Figure S11).”

- 136: Citations needed.
 - The citation for this statement is the same as the one for the end of the sentence (Sanjuán *et al.* 2010); we have duplicated it within the sentence to avoid confusion.
- 152: Ref 28 is not a super-tree paper (it is the Fritz et al 2009 paper). Which super-tree did you use? Suggest checking other citations.
 - This is the correct citation; Fritz et al created a supertree for this paper, which we used for the analysis, as in a previously published analysis (Olival *et al.* 2017).
- 534: Patristic distance?
 - We have clarified that this was the patristic distance.

Best,

Nick Fountain-Jones

References

- Davies, T.J. & Pedersen, A.B. (2008). Phylogeny and geography predict pathogen community similarity in wild primates and humans. *Proc. R. Soc. B Biol. Sci.*, 275, 1695–1701.
- Huang, S., Bininda-Emonds, O.R.P., Stephens, P.R., Gittleman, J.L. & Altizer, S. (2014). Phylogenetically related and ecologically similar carnivores harbour similar parasite assemblages. *J. Anim. Ecol.*, 83, 671–680.
- Olival, K.J., Hosseini, P.R., Zambrana-torrel, C., Ross, N., Bogich, T.L. & Daszak, P. (2017). Host and viral traits predict zoonotic spillover from mammals. *Nature*, 546, 646–650.
- Rushmore, J., Caillaud, D., Matamba, L., Stumpf, R.M., Borgatti, S.P. & Altizer, S. (2013). Social network analysis of wild chimpanzees provides insights for predicting infectious disease risk. *J. Anim. Ecol.*, 82, 976–986.
- Sanjuán, R., Nebot, M.R., Chirico, N., Mansky, L.M. & Belshaw, R. (2010). Viral mutation rates. *J. Virol.*, 84, 9733–9748.
- Stephens, P.R., Altizer, S., Ezenwa, V.O., Gittleman, J.L., Moan, E., Han, B., *et al.* (2019). Parasite sharing in wild ungulates and their predators: Effects of phylogeny, range overlap, and trophic links. *J. Anim. Ecol.*, 88, 1017–1028.
- Wells, K., Gibson, D.I., Clark, N.J., Ribas, A., Morand, S. & McCallum, H.I. (2018). Global spread of helminth parasites at the human-domestic animal-wildlife interface. *Glob. Chang. Biol.*, 24, 3254–3265.
- Wells, K., Morand, S., Wardeh, M. & Baylis, M. (2020). Distinct spread of DNA and RNA

viruses among mammals amid prominent role of domestic species. *Glob. Ecol. Biogeogr.*, 29, 470–481.

Willoughby, A.R., Phelps, K.L., Predict Consortium, predict@ucdavis.edu & Olival, K.J. (2017). A comparative analysis of viral richness and viral sharing in cave-roosting bats. *Diversity*, 9, 1–16.

Reviewers' Comments:

Reviewer #4:

Remarks to the Author:

The authors have done a good job responding to my comments. I still think they should include another analysis approach to see if this supports the results presented, but I will leave it to the editor to make the call. Otherwise, the manuscript is an excellent contribution to the literature - well done.